# Reduced Cell Adhesion on LightPLAS-Coated Implant Surfaces in a Three-Dimensional Bioreactor System

**DOI:** 10.3390/ijms241411608

**Published:** 2023-07-18

**Authors:** Kai Oliver Böker, Linda Gätjen, Christopher Dölle, Katarina Vasic, Shahed Taheri, Wolfgang Lehmann, Arndt Friedrich Schilling

**Affiliations:** 1Department of Trauma Surgery, Orthopaedics and Plastic Surgery, University Medical Center Goettingen, Georg-August-University, 37075 Goettingen, Germany; 2Fraunhofer Institute for Manufacturing Technology and Advanced Materials IFAM, 28359 Bremen, Germanychristopher.doelle@ifam.fraunhofer.de (C.D.)

**Keywords:** 3D bioreactor system, cell adhesion, surface manipulation, surface coating, medical implant modification

## Abstract

Most implants used in trauma surgery are made of steel and remain inside the body only temporarily. The strong tissue interaction of such implants sometimes creates problems with their explantation. Modified implant surfaces, which decrease tissue attachment, might allow an easier removal and therefore a better outcome. Such a modification must retain the implant function, and needs to be biocompatible and cost-effective. Here, we used a novel VUV-light (Vacuum-Ultraviolett)-based coating technology (LightPLAS) to generate coated stainless-steel plates. The tested LightPLAS coating only had an average thickness of around 335 nm, making it unlikely to interfere with implant function. The coated plates showed good biocompatibility according to ISO 10993-5 and ISO 10993-12, and reduced cell adhesion after four different time points in a 2D cell culture system with osteoblast-like MG-63 cells. Furthermore, we could show decreased cell adhesion in our 3D cell culture system, which mimics the fluid flow above the implant materials as commonly present in the in vivo environment. This new method of surface coating could offer extended options to design implant surfaces for trauma surgery to reduce cell adhesion and implant ingrowth. This may allow for a faster removal time, resulting in shorter overall operation times, thereby reducing costs and complication rates and increasing patient wellbeing.

## 1. Introduction

An important early step in fracture healing is the immobilisation of the fractured bone with metal implants, which provide the necessary support and help the bone to heal. There are several ways to immobilise broken bones in the correct position, including screws, plates, Kirschner wires, cerclage wires or external/internal fixators and intramedullary nails [1,2,3]. 

In the case of bone implants, the current scientific focus is on improving the cell adhesion of human osteoblasts. The aim is to force the ingrowth of implants that remain permanently in the body and to establish the highest possible adhesion of the implant material to the bone tissue, e.g., dental implants [4] or endoprostheses [5]. For this purpose, roughened surfaces, e.g., by means of corundum blasting coatings (e.g., Dot BONIT^®^ [6]) or a PVD coating with titanium nitride, titanium niobium nitride or zirconium nitride (ZrN), are used [7,8]. This is particularly useful for hip implants, where high incorporation inside the human bone is necessary. Femoral stem prostheses have undergone many modifications and design optimizations since their introduction and many studies have focused on the improved integration of these implants, since they remain inside the body for a extended time [9].

In contrast, in the osteosynthetic treatment of bone fractures, trauma implants remain in the body only temporarily [10]. This is particularly true for facial and extremity implants, which are often perceived as uncomfortable by the patient [11], especially in growing children. The removal of such implants can be complicated by the ingrowth of musculoskeletal tissue, which can obstruct the surgeon’s view and make it difficult to access the implant. Therefore, inhibition of cell adhesion may simplify the surgical removal process. This would likely result in shorter operation times with lower complication rates, such as smaller wounds, reduced infection rates, reduced risk of soft tissue damage, faster wound healing and less pain overall.

### 1.1. Implants

Currently, two materials are preferred for trauma implants: medical stainless steel and titanium. In Europe, the use of titanium (oxide) material dominates, whereas in the USA, stainless steel tends to be used [12]. Currently, stainless steel is the most widely used implant material worldwide due to its low material cost and excellent biomechanical properties [12,13]. Stainless steel is significantly stiffer than bone and has been shown to promote bone healing [14]. It is less expensive than comparable implant materials, such as titanium, and offers an easy-to-polish surface [15]. It contains chromium oxide (Cr_2_O_3_), which forms a strong and adherent layer on the surface of the implant, which prevents corrosion [16].

Anodization is the preferred finishing process for titanium implants. In this process, the natural oxide film is replaced by a thick oxide layer in an electrolytic bath with a strong alkaline pickling agent. Type II anodizing creates a uniform surface condition and modifies the biological and biomechanical properties. Type III anodizing adds a thin titanium dioxide film for colour classification (for example the DOTIZE^®^ process [17]). The DOTIZE process reduces protein adsorption on the implant surface by up to 19%. Anodizing medical stainless steel is difficult due to its higher conductivity, especially its iron content, which limits current and slows oxide layer growth, making it uneconomical. In this project we used the LightPLAS approach, changing the surface properties of medical steel to modify cell adhesion, which is described in detail in 1.3.

### 1.2. Implant Removal

Once the fracture has healed, most metal implants need to be removed. Hardware removal is one of the most common surgical procedures performed worldwide [18]. In Germany alone 180,000 implants were surgically removed in 2010, excluding removals performed by general practitioners [19]. Implant removal is a critical procedure, with the risk of wound infection, peripheral nerve injury and refracture [20]. The integration of implants into the surrounding soft and hard tissue amplifies the problem. When steel implants integrate into bone, they are typically covered by a thin fibrous surface. If the attractiveness of stainless steel can be increased by reducing cell adhesion and thus increasing the rate of use, the lower material costs of stainless steel compared to titanium can result in a cost advantage for the healthcare system. Depending on material quality, stainless steel is 2–4 times less expensive than titanium [21]. Over the past decade, there has been an increase in the use of implants to stabilize fractures, with a corresponding increase in the need for implant removal [19]. This situation increases the cost of treating bone fractures; for example, the total cost of treating bone fractures in Germany in 2016 was approximately EUR 250 million per year [22], and the numbers are expected to increase in the future. Furthermore, the need for surgical implant removal in the Western world has increased in recent years and will continue to do so, underscoring the overall importance of implant removal with reduced complications. Last but not least, due to demographic changes and modern sports trends, the number of fractures in all age groups is expected to continue to increase [22]. A possible approach to reduce the overall costs, operation times and complication rates could be to prevent material incorporation, for example by using LightPLAS-coated implants.

### 1.3. LightPLAS Coating of Steel Implants

The applied LightPLAS technology describes a light-based coating process for surface functionalisation that is suitable for various surface materials, especially when a biocompatible coating is required [23,24,25]. The process consists of two basic steps. In the first step, a precursor is applied to the surface as a thin liquid film. In the second step, high-energy photons are used to cross-link the liquid film into a solid, cross-linked layer. These two steps are flanked by pre-treatment procedures such as cleaning, measures to homogenise the liquid film or a post-treatment step, such as the grafting step used in this work. Typically, the functional layers produced by LightPLAS are thinner than 500 nm. They are therefore significantly thinner and more true-to-scale than paint layers, which typically have thicknesses of more than 5 µm. Another advantage is the absence of the reactive chemistry characteristic of paint or sol–gel coatings, e.g., photoinitiators in UV coatings. The LightPLAS precursor used here, on the other hand, can be completely chemically inert, making it suitable for use in medical devices, particularly in vivo applications such as implants. Cross-linking is achieved using 172 nm radiation from an excimer lamp, which is technically equivalent to a dielectric barrier discharge [26]. Due to its high photon energy, the light is able to break the molecular bonds of the precursor [27,28,29]. Reactive fragments are generated that interact with each other and with a three-dimensional cross-linked layer. In this work, a non-reactive linear silicone oil (AK50) is used as a precursor, which is converted to an inorganic glass-like layer by irradiation. The layer chemistry is very similar to that of a plasma polymerisation coating, which is deposited from the gas phase rather than from a liquid [30].

### 1.4. Hypothesis

In this project, we will compare the effect of coated and uncoated medical steel on bone cell adhesion, cell viability and cell morphology. We hypothesise that a low-energy coating with hydrophobic properties will decrease cell adhesion by inhibiting the interaction with the metal surface. In addition, it is expected that the reduced cell adhesion will lead to an increased number of rounder cell phenotypes. If biocompatibility can be demonstrated, no impact on cell viability is expected.

## 2. Results

Figure 1 shows an uncoated stainless-steel reference sample (A) and a coated sample (B) in plain view. The coating can be identified by the yellow-orange interference colour. The surrounding edge is uncoated due to a fixture used during the coating process to prevent back and edge coating. The bluish surrounding edge merely reflects the gradual increase in coating thickness. Only the central area of the sample was considered for the cell studies. The slightly different cloud-like colouration in the central area indicates small differences in the layer thickness. The average film thickness here is 335 nm and the film thickness deviation is about 1%. Figure 1C shows a cross-section of the film thickness. The water contact angle measurements (Figure 1D,E) show a smaller contact angle for uncoated samples of about 75° ± 7° (n = 3), while the coating increased the contact angle to more than 100, 6° ± 10° (n = 9). The high angle illustrates the effect of grafting with HMDSO, which results in hydrophobic properties. In contrast, without grafting, the angle immediately after cross-linking is less than 10°, indicating hydrophilic properties. The polar fraction of the surface energy was <0.9 mN/m ± 0.4 mN/m (n = 9) for the grafted coatings and 39 mN/m ± 0.4 mN/m (n = 9) for the ungrafted coatings. According to the XPS surface elemental analysis, the top layer of the grafted coating consists of carbon C(1s): 22.2–22.9%, oxygen O(1s): 53.2–54.4% and silicon Si(2p): 23.4–23.8%. After cross-linking, but without grafting, the composition is C(1s): 10–15%, oxygen O(1s): 57–65% and silicon Si(2p): 25–28%. The cross-cut test for coating adhesion was GT0, i.e., no flaking or delamination was visible near the scratches after peeling off the tape.

Appendix A shows the IR spectra of the non-cross-linked liquid layer after the homogenisation step and the final LightPLAS coating in comparison. Two individual measurements are shown in the wavenumber range from 500 to 4000 cm^−1^ for each of the two variants, which are almost identical. The absorbance is given in arbitrary units.

The biocompatibility studies showed no cytotoxic effects of the uncoated or LightPLAS-coated steel plates. Compared to the reference cells on plastic surfaces, the cells on uncoated steel plates had 89% and cells on coated steel plates had 93 % cell viability. HEMA treatment reduced the cell viability to 0% (Figure 2).

A typical procedure for medical devices is sterilization using gamma radiation before being placed on the market. This step is usually performed after packaging the product. Gamma irradiation in an air atmosphere typically produces reactive oxygen species. Therefore, there is a risk that the oxygen radicals formed will interact with the LightPLAS coating and destroy its hydrophobic character. To minimise these unwanted effects, the samples were first sealed in aluminium foil under vacuum. The coated samples were then sterilised by gamma irradiation at 30 kGy. The evaluation was carried out using the water contact angle and showed no changes compared to the coated reference with angles around 100°. In addition, the coating showed no optical changes or damage.

For the analysis of cell morphology, circularity and cell area, fluorescence images were analysed via ImageJ due to the increased contrast compared to phase-contrast pictures (Figure 3A,B). The threshold parameters were set accordingly to generate a cell mask (Figure 3C–J). With this system, it was possible to analyse 14,500 cells on the reference plates (uncoated) and 6000 cells on the coated plates.

In a first step, the cellular influence of the coated metal surfaces was analysed in a standard 2D in vitro setup at four different time points (3–72 h, Figure 4). For all samples, the number of cells increased with the duration of storage (Figure 4A). Cells on the reference plates grew from approx. 20 cells after 3 h to approx. 50 cells after 72 h of incubation. The coated surfaces showed a reduced cell count from the beginning (approx. 10 cells after 3 h). After 3 days of culture, the cell number was increased to approx. 22 cells. At all time points, a significant decrease in cell number was detected on coated metal surfaces. Cell circularity was increased on coated metal surfaces compared to non-coated surfaces up to 24 h, while no differences in circularity were observed after 48 and 72 h (Figure 4B). On the other hand, the single-cell area was decreased on coated surfaces up to 24 h, while no differences in cell area were observed for 48 h and 72 h (Figure 4C).

In the second step, MG-63-RFP cells were analysed under dynamic conditions with medium flow in a 3D bioreactor setup. The bioreactor was designed with CAD software (Figure 5A) and 3D-printed (Figure 5B). The medium flow system including an Ibidi pump and the valve/tube system is depicted in Figure 5C. The cross-section of the system (Figure 5D) shows the metal plate with fluorescent cells and medium flow on top.

MG-63-RFP cells were analysed with two different flow conditions, 5 mL/min (low medium flow, Figure 6) and 10 mL/min (defined as high medium flow, Figure 7), at two different time points (24 h, Figure 6A and Figure 7A, and 48 h, Figure 6B and Figure 7B).

At low flow conditions, a reduced cell count was observed after 24 h (51%) and 48 h (56%). Circularity was significantly increased after 48 h, while no change was observed at the earlier time point. For the single-cell area, no changes were detected. At high flow conditions, we could also see a cell count reduction of 67% after 24 h and 68% after 48 h, while circularity was only increased after 24 h. Furthermore, the single-cell area was reduced at this time point, while no changes were observed after 48 h. Mean values with SD are summarized in Table 1.

## 3. Discussion

In this project, the cellular interaction of human osteoblast-like cells with surface-modified implants was analysed. For the surface modification, a novel technique (LightPLAS) was used to theoretically reduce cell adhesion and thus facilitate implant removal.

In the first step, the uniformity of the coating layer’s thickness was analysed. The average layer thickness is mainly controlled by the aerosol exposure time. Due to the partially turbulent gas flows in the aerosol chamber, the deviation in the reproducibility of the thickness was about 10%. For future industrial implementation, it is certainly advisable to increase the reproducibility. However, no significant changes in surface properties are expected, as shown by previous studies on the cell attachment of MG-63 osteoblasts on thinner coatings [31]. Instead, the local uniformity of the coating is more important. The liquid precursor coating is deposited as a droplet distribution from an aerosol. As a result, there are initially large differences between partially uncovered surface areas and areas with a thick coating due to the superposition of multiple droplets. Uncoated areas will not show cell-adhesion-reducing properties, and thick precursor layers may not be sufficiently cross-linked by the VUV radiation. As shown in Figure 1B, heat treatment with IR radiation is a good technique to even out the liquid coverage. Alternative techniques for precursor application would be, for example, spray application, which is used for paint, or dipping. Spraying also uses droplets, but typically produces a film thickness of a few microns. Submicron coatings require large amounts of diluents. Aerosol application, on the other hand, does not require diluents. Although dip processes can produce very uniform and thin films, solvents are still required. Dipping is ideal for flat surfaces such as glass, but for 3D geometries such as typical implant forms, cavities and edges will cause localized imperfections. Suitable 3D thin functional coatings with a high degree of uniformity can be achieved with plasma technology. In the medical field, a number of applications of plasma functional surfaces based on hexamethyldisiloxane with hydrophilic or hydrophobic and, above all, cell-growth-supporting properties are already known [32,33,34,35]. However, the need for low-pressure technology makes plasma functionalization significantly more expensive than LightPLAS.

In Appendix A, the effect of UV cross-linking is clearly visible when looking at the absorption bands in the IR spectra. There are several wavenumber regions where the effect of UV cross-linking is clearly visible. Of most importance is the peak of maximum absorbance of the non-cross-linked silicone oil reference spectra that is located at 1111 cm^−1^ with a broad shoulder around 1040 cm^−1^. It can be assigned as an asymmetric Si-O stretching vibration. By UV irradiation, the maximum absorbance is typically shifting to higher wavenumbers. At the same time the shoulder decreases in intensity. A further peak of relevance of the non-cross-linked silicone oil can be seen at 1268 cm^−1^, which is assigned as symmetric bending vibration δs(CH_3_) of the methyl groups attached to Si. By UV irradiation, the symmetric bending vibration is shifted towards 1280 cm^−1^. Additionally, the band absorbance decreases in intensity. The shift of the δs(CH_3_) vibration indicates a change in the structural unit from -O-Si(CH_3_)_2_-O- to -O-Si(CH_3_)(O-)_2_ caused by cross-linkage of chains as well as by the replacement of methyl by hydroxyl groups. All observed changes in the spectra are in agreement with the data found in the literature, where further details on the different behaviours of the absorption bands can be found [27,28,29,36,37]

Looking at the low surface energy of 21–35 mN/m and the high water contact angle of above 90° for the LightPLAS coating, the XPS elemental composition seems confusing because of the low amount of carbon detected. According to the XPS analysis, the surface should have a more hydrophilic character than the non-grafted coating. One possible reason may be the depth of information in the XPS analysis, which is about 10 nm, and its sensitivity decreases exponentially with depth. The XPS measurement registers, at least proportionally, the properties of the ungrafted coating, whose water contact angle, with values below 10°, is much more consistent with the elemental analysis. In contrast, the water droplet responds only to the interfacial properties of the coating, which are determined by the grafting. A detailed analysis suggests that grafting has deposited an additional layer with an average thickness of about 1 nm. Due to the small thickness compared to the LightPLAS coating with more than 300 nm, it is not expected that the grafting produces a significant change in the IR spectrum, as the results of the IRRAS analysis always correspond to an average over the whole layer. However, a detailed characterization of the grafting is necessary to understanding the interaction in detail. As shown, this property is maintained when the presence of air in the immediate vicinity of the grafted surface can be excluded during gamma irradiation. This result has implications for the future treatment of implant surfaces during gamma sterilization. Due to the low surface energy, the LightPLAS coating is defined as hydrophobic. Instead, hydrophilic surface properties are known to be required for increased bonding between biological cells and implant material [38]. Corresponding evidence regarding protein adsorption and cell adhesion can be found increasingly in the literature [39,40,41]. For example, a chitosan coating shows a water contact angle of 76.4°, and an uncoated titanium plate around 32.2° [42]. Plasma-treated samples with contact angles below 10° show potential to promote osteoblast attachment [40]. Furthermore, time-dependent effects reducing the surface energy are reported as disadvantageous for osseointegration [42,43]. In conclusion, high contact angles or low surface energies are unsuitable for optimized protein adhesion as they suggest the potential for reduced cell adhesion instead.

A biocompatibility test was performed to evaluate the suitability of the modified surfaces for clinical use. Both the coated and unmodified medical steel plates showed no evidence of cytotoxicity. All cell viabilities observed were above 70%, which is defined as biocompatible according to DIN EN ISO 10993-5.

The cell adhesion of human MG-63 cells was significantly reduced by the LightPLAS coating in a standard 2D environment after between 3 and 72 h of culture. Furthermore, increased circularity was observed at up to 1 day, while the cell area was reduced during this time period. Reduced cell numbers combined with increased circularity were also observed with other surface modifications, such as Hydromer’s F200™ and F202™ polymer formulations [44]. Murine L929 fibroblasts were analysed on F200 or F202 coated and uncoated steel plates and a reduced cell number as well as a rounded shape were observed. Similar results were obtained by a group studying the adhesion of murine fibroblasts to titanium alloys [45], showing reduced cell adhesion on different metal types for osteosynthesis material.

In general, cell adhesion is influenced by surface proteins called cell-adhesion molecules (CAMs), which can be further categorized into the immunoglobulin superfamily, cadherins, integrins and selectins [46]. Cell junctions are formed by an interaction with these proteins and their ligands [47]. However, these interactions usually occur between neighbouring cells or between cells and the extracellular matrix. Pure metal surfaces do not provide protein interactions per se. Therefore, cells adhere to metal surfaces via focal contacts and extracellular matrix adsorption [45,48]. Furthermore, CAMs can cluster to focal adhesions [49], which have a sub-µm lateral dimension [45], while the spikes at these structures interact with the surface. For example, oxidation can change surface properties and influence the interaction with CAMs and therefore reduce cell adhesion [50].

Typically, cell adhesion to metal surfaces is tested in a 2D in vitro setup as described above. To mimic more closely the in vivo situation, we established our 3D bioreactor system with fluid flow around the metal surfaces including living cells.

Since metal surfaces are not transparent to visible light, phase-contrast microscopy is not possible on these surfaces. Due to the stable lentiviral transduction of the osteoblast cells, a permanent RFP signal is visible and no additional staining, including cell fixation, is required. This technique allows live-cell imaging on metal plates, which is unique in this experimental setup. In our bioreactor setup, even stronger effects were observed compared to the 2D cell culture setup. Especially under high flow conditions, cell number was significantly reduced by 56%, while circularity was significantly increased after 48 h on coated surfaces. These results in a 3D environment support the observations in a standard 2D environment. Nevertheless, the mechanism behind the reduced cell adhesion remains unclear. Future studies should focus on the detailed analysis of surface proteins and their distribution on the cell surface. This analysis could be performed by fluorescence staining and high-resolution microscopy.

All together, we compared the effect of coated and uncoated medical steel on bone cell adhesion, cell viability and cell morphology. Our hypothesis that a low-energy surface coating decreases cell adhesion and leads to a rounder cell phenotype without cytotoxic effects was accepted.

In general, cell adhesion and osseointegration are important processes for medical osteosynthesis implantations [51]. In load-bearing regions, osseointegration is crucial for the stability of the implant. This is one reason why studies focus on the improvement of osseointegration, in contrast to this study [52,53,54]. However, many studies describe complications during intramedullary nail removal or the need for special instruments [55,56,57]. Therefore, in recent decades, some studies have also focused on reducing osseointegration, like Levanus et al., who showed a decreased osteocalcin and collagen 1 expression after cultivation on Ti300 substrates, a titanium surface with 300 nm pores [58]. Furthermore, polymer-based coatings, like Hydromer’s polymeric formulations F200™ and F202™, were used to reduce cell adhesion [44]. Other workgroups focused on tendon adhesion on titanium surfaces using Dotize^®^ or plasma electrolytic oxidation (PEO) modifications. Dotize^®^ replaces the natural oxide film on titanium implants with a thicker oxide coating. Dotize^®^ and PEO-03 showed significant reductions in tendon adhesion of 55.1% and 64.3% [17].

Another method of modifying the surfaces of titanium alloys is using passive films produced by potentiostatic polarization [59]. Ti–6Al–4V is widely used and passive films lead to a higher corrosion resistance, which is particularly interesting for the biomedical industry. Furthermore, titanium alloys can be modified by selective laser melting and triply periodic minimal surface lattice, producing high-entropy alloys with a Young’s modulus (6.71–16.21 GPa) very close to human trabecular bone.

Since reduced cell adhesion seems to be the most important for intramedullary nails, further research should focus on curved or round surfaces. The removal of intramedullary nails can be difficult, while the use of LightPLAS-coated implant materials can simplify the process. The localized inhibition of cell adhesion leads to reduced integration of the implant into the surrounding tissue, while reduced ingrowth potentially leads to reduced removal time and therefore reduced cost to the healthcare system as well as increased patient comfort. However, the modification of these curved surfaces seems to be more complicated, since the distance of the material to the aerosol source and the VUV lamp is not fixed as in the case of flat metal plates. This methodological challenge should be addressed in future experiments.

Furthermore, many studies on modified surfaces focus on biofilm prevention or bacterial growth inhibition [60,61,62,63]. Since antibiotic treatment for infected implant material is limited, these modifications are highly clinically relevant and should be the focus of further research. In addition, the stability of the coated surface is an important factor. In this study we did not observe a peeling of the surface; nevertheless, scratches were observed after cell culture handling. Future studies will focus on implant and surface stability using surface analysis and biomechanics after handling. For the clinical use of such modified implants, animal models are essential to analyse implant ingrowth, implant stability during motion, immune response and the possible influence on bone healing, which will be considered in future studies.

## 4. Material and Methods

### 4.1. Study Overview

The design of this study including the different analysis types and experiments are summarized in Figure 8.

### 4.2. Coating Procedure

Circle-like 316 L alloy medical stainless-steel samples were used for the coating experiments. The surface was polished and possessed specular gloss with a roughness of ra ~10 nm comparable to medical steel trauma implants. In the first step, the surfaces were cleaned with methyl ethyl ketone (C_4_H_8_O) in an ultrasonic basin and subsequently with isopropanol (C_3_H_8_O) and activated by vacuum ultraviolet radiation (central wavelength at 172 nm, type Xeradex^®^, Radium Lampenwerk Wipperfürth, Wipperfürth, Germany) in air atmosphere at 244 mJ/cm^2^. Due to the incorporation of oxygen into the surface, the initial surface possessed a surface energy around 70 mN/m. In the second step, a thin liquid layer of a non-functionalized linear silicone oil (AK50, Wacker Chemie AG, Munich, Germany) was applied as a precursor to the steel surface by aerosol application. The application with the aerosol took place in a closed chamber at normal pressure. The aerosol itself was generated by an aerosol generator with a baffle plate. The average droplet size was approximately 3–5 µm. The stainless-steel samples showed an inhomogeneous layer thickness typical of droplet distribution. For thickness homogenization, the liquid covering was irradiated with an infrared lamp at a temperature of 100 °C in air for five minutes in the third processing step. In the subsequent step, the liquid covering was cross-linked under oxygen-reduced atmosphere (approx. 2% O_2_, rest N_2_) in a closed chamber with VUV radiation from a KrF-excimer lamp (Xeradex^®^) with a radiation dose of 5.6 J/cm^2^ with static positioning between lamp and sample. During cross-linking, a constant gas flow of 30,000 sccm was injected into the approximately 125 L chamber locally at the sample position. Since ozone is generated by VUV radiation when oxygen is present and accumulates in a static atmosphere, the gas flow provided approximately constant conditions at the sample position throughout the irradiation time. Finally, after opening the chamber and brief contact with air atmosphere, the irradiated layer was grafted with HMDSO (hexamethyldisiloxane, Wacker Chemie AG) in step five. To minimize short-term changes in the layer chemistry, e.g., due to dynamic rehydrophobization processes, the coated samples were treated for artificial aging for 30 min at 100 °C in the oven. Finally, the samples were irradiated using a 254 nm UV lamp (uv-technik Speziallampen GmbH, Ilmenau, Germany) for surface disinfection prior to cell assays. The complete coating workflow is summarized in Figure 9.

### 4.3. Contact Angle Measurement

A common method for characterization of the surface properties is contact angle measurement. The contact angle contains the feedback from the upper interface of the coating to the surrounding air and a liquid drop. Typically, the progressive contact angle is determined for three different types of liquids, namely water, diodmenthane and ethylene glycol. A liquid pending drop is applied to the surface. An image of the drop is recorded by camera and evaluated by a shape-analysing software. The evaluation method according to Owens–Wendt–Rabel–Kaelble allows the differentiation between disperse and polar surface energy fractions. In this work, a commercial system from DataPhysics, type OCA 50, was used. The contact angles given typically represent the average of a series of 50 measurement points for each liquid. At least 3 biological replicates (*n* ≥ 3) were measured for coated and uncoated samples.

### 4.4. Layer Thickness Detection

Reflectometric thin-film measurement is suitable for measuring the thickness of thin, transparent coatings. It is based on the principle of thin-film interference. The reflection spectrum of a sample is recorded over the widest possible wavelength range. Knowing the refractive index for the layer material, a theoretical reflection spectrum can be calculated for a given layer thickness. The measured spectrum and model are matched by varying the model layer thickness, and thus the most probable layer thickness is determined by optimal matching both spectra. In our case, the reflectometer (NanoCalc) was adapted to the profilometer PLu neox (Schäfer Technologie GmbH, Langen, Germany) with a glass fibre. The measuring spot size was 7 µm. The coating material was assumed to be thermal SiO_2_ and stainless steel as highly reflective base material. By moving the coated sample, a spatially resolved lateral section through the coating layer can be recorded.

### 4.5. Surface Elemental Composition by X-ray Photoelectron Spectroscopy (XPS)

XPS is a widely used technique for surface-sensitive, quantitative analysis of surface chemistry with respect to elemental composition, except for hydrogen and helium. Here, XPS measurements for the elemental composition were performed using a KratosAxisUltra Spectrometer, Kratos Analytical Corp. The high detection sensitivity of the method is element-specific and is about 1000 ppm. The typical depth of information is approximately 10 nm. Reported values represent an average of two different surface spots of the same sample.

### 4.6. Coating Bond Strength

Due to the low film thickness and the hydrophobic properties of the LightPLAS coating, only a qualitative test of the film’s adhesion to the titanium substrate could be carried out. This cross-cut test was carried out in accordance with DIN EN ISO 2409 to evaluate adhesion of paint. A crossed grid of 6 × 6 lines with a spacing of 1 mm or 25 individual fields was manually scratched into the surface of the coating down to the stainless-steel base material using a scalpel. The surface was then activated by VUV light to enable adhesion of an adhesive tape (tesa^®^ fabric tape 4651). The tape was peeled off and the remaining layer or the spalling near the scribe lines were evaluated. The number of investigated samples was three.

### 4.7. IR Spectroscopy

We employed infrared spectroscopy for surface composition analysis. The absorption bands exhibited characteristic chemical binding ratios, making the VUV cross-linking of the liquid silicone oil clearly visible. Infrared spectra in IRRAS (Infrared Reflection–Absorption Spectroscopy) configuration were obtained at an 45° angle of incidence relative to the surface normal. A single-beam Fourier transform spectrometer, specifically the VERTEX 80 model from Bruker Optik GmbH, was used. The spectrometer featured a liquid nitrogen-cooled MTC detector and a resolution of 2 cm^−1^. To ensure precise results, we performed 128 scans and subsequently subtracted the resulting spectrum from a background spectrum obtained using a non-coated stainless-steel sample as a reference. We compared the measurements of the liquid silicone oil layer after homogenization, which corresponds to step three of the coating procedure, with the final LightPLAS coating. Each analysis involved two samples (*n* = 2).

### 4.8. Cell Culture

Human osteoblast-like MG-63 cells and mouse fibroblasts L929 were purchased from Cell Lines Service (CLS, Eppelheim, Germany). L929 cells were cultured in Roswell RPMI 1640 Medium (Gibco, Waltham, MA, USA) including L-glutamine, 10% FCS and 1% antibiotics (penicillin and streptomycin), while MG-63 cells were maintained in DMEM Ham’s F12 medium (Gibco, Waltham, MA, USA) with 10% FCS and 1% antibiotics (penicillin and streptomycin) according to standard conditions in a humidified incubator at 37 °C and 5% CO_2_.

### 4.9. Biocompatibility Test

The biocompatibility was assayed according to the DIN EN ISO 10993-5 and -12 guidelines using the cleavage of the tetrazolium salt WST-1 (4-(3-(4-Iodophenyl)-2-(4-nitrophenyl) -2H-5-tetrazolio)-1,3-benzene disulfonate) to formazan. Ten independently manufactured samples with LightPLAS coating were each tested in triplicate (n = 10). The samples were first washed with 1 mL RPMI 1640 medium. Then, the extraction was carried out in 1 mL RPMI 1640 medium per sample at 37 °C, 72 h prior to the application of extracts on cells. L929 cells (100 µL cell suspension, 10^5^ cells/mL) were seeded into a 96-well plate and incubated for 24 h at 37 °C, 5% CO_2_. An amount of 100 µL supernatant was substituted with 100 µL extract. After an additional 24 h of incubation, the solutions were removed from the wells and substituted with 150 µL of the proliferation reactant WST-1 (10 µL WST-1/mL RPMI). After 1 h of incubation with WST-1, 100 µL of each well was transferred to a new plate and the absorbance was measured at 450 nm with a multimode reader (Mithras LB 940, Berthold Technologies, Bad Wildbad, Germany). Untreated cells were used as negative control (biocompatible) and a treatment with 10% hydroxyethyl methacrylate (HEMA) was used as positive control (cytotoxic).

### 4.10. Lentivirus Production

Vesicular stomatitis virus glycoprotein (VSV-G) pseudotyped lentiviruses were generated in HEK293FT cells. Cells were seeded in 6-well plates at 1 × 10^6^ cells/mL and directly transfected with pLemir-NS [64] for RFP delivery, psPAX2 for viral capsid proteins and pCMV-VSV-G [65] for viral envelope proteins (summarized plasmids are listed in Table 2). After 24 h, medium was collected every 24 h for 120 h in total. After 5 days, medium was centrifuged for 30 min at 2000× *g*, filtered through 450 nm filters (Sartorius, Göttingen, Germany) and concentrated in Vivaspin columns (Sartorius, Göttingen, Germany).

### 4.11. Generation of Stable RFP-Expressing MG-63 Cells

MG-63 cells were seeded in 24-well plates (50,000 cells/well). After 24 h, the cells were infected with lentivirus carrying VSVG as an envelope protein and RFP as a gene of interest. After, 3 passages cells were sorted by fluorescence-activated cell sorting (FACS). Therefore, 500,000 cells were harvested and sorted via BD FACS ARIA II instrument and the red filter set (633 nm). RFP-positive cells (MG-63-RFP) were selected, sorted and further cultured as described above.

### 4.12. Morphological Analysis

MG-63-RFPs were seeded on uncoated and coated surgical steel plates inside a 12 well plate. Images were obtained with Leica DMi 8 microscope 3, 24, 48 and 72 h after seeding. Analysis of cell number, size and circularity was performed using ImageJ. Threshold for detecting cells was set between 65 and 255 and adjusted for each picture. Circularity describes cell shape, while a value of 1 corresponds to an ideal circle. Single-cell size was measured in µm^2^. In total, 20 samples on untreated plates (*n* = 20) and 20 samples on coated plates (*n* = 20) were analysed. For each biological replicate, six single pictures were analysed. Six pictures were taken at the same spot for all replicates. The impact of the coating was evaluated via *t*-test (GraphPad Prism 5, La Jolla, CA, USA). *p*-values < 0.05 were considered significant. Data are presented as mean values and standard deviations (SDs).

### 4.13. Bioreactor Setup

MG-63-RFP cells were seeded on uncoated and coated surgical steel plates inside a 24-well plate. After 3 h, cells on reference or coated metal plates were transferred into the bioreactor setup system. The Ibidi pump was attached, and two different flow conditions (5 mL/min and 10 mL/min) were set up in the software. Every Ibidi pump supplied two different bioreactors, one with a reference metal plate and the other one with a coated metal plate. Six individual pictures were taken at two different time points (24 h and 48 h) for every flow condition. In total, ten biological replicates (*n* = 10) were measured for reference and coated steel plates. Images were analysed using Image J (see above). The impact of the coating was evaluated via *t*-test (GraphPad Prism 5, La Jolla, CA, USA). *p*-values < 0.05 were considered significant. Data are presented as mean values and standard deviations (SDs).

## 5. Conclusions

In conclusion, our results demonstrate a novel way to modify implant surfaces using a VUV-light-based coating. This surface modification with a thickness of approx. 330 nm leads to a reduced cell adhesion of human osteoblast-like cells in standard 2D in vitro environments (significant reduction of up to 65% for all measured time points), but also under dynamic flow conditions in a 3D cell culture system (significant reduction of 56% for 5 mL/min and 68% for 10 mL/min). Furthermore, we were able to demonstrate increased cell circularity (up to 10%) and reduced cell area (up to 21%) under flow conditions using LightPLAS coatings (Table 1). These results are based on the low-energy layer properties of the LightPLAS coating on medical steel. It is assumed that similar properties can also be achieved after coating other materials such as titanium. However, this would need to be demonstrated. The LightPLAS technique may be a suitable method to reduce the local osseointegration of osteosynthetic materials such as medical steel for easier removal, reduced surgical and healthcare costs and improved patient welfare.

## Figures and Tables

**Figure 1 ijms-24-11608-f001:**
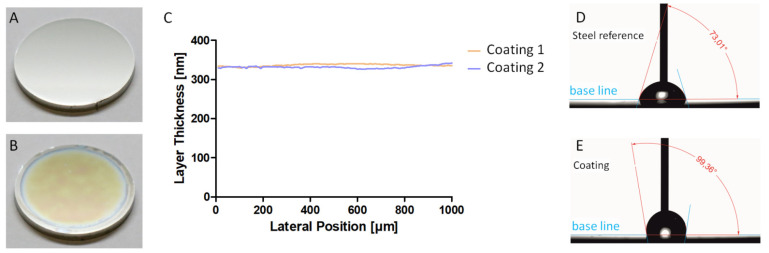
Coating result, layer thickness and surface properties. Comparison of the uncoated medical stainless-steel reference (**A**) and coated (**B**) samples. The coating is noticeable from interference colours due to its low layer thickness of 335 ± 3 nm and 332 ± 3 nm, respectively (**C**). The surrounding edge is uncoated. Measurement of the water contact angle yield around 75° ± 7° for the reference (**D**) and 100° ± 10° for the coated sample (**E**).

**Figure 2 ijms-24-11608-f002:**
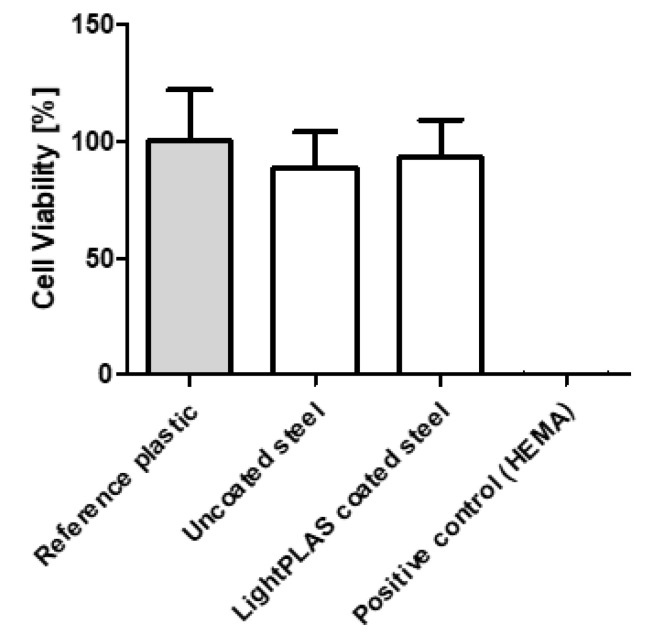
Biocompatibility test for the LightPLAS-coated implants. Based on WST-1 assay according to DIN EN ISO 10993-5 and 10993-12, ten independently manufactured samples with coating, uncoated samples, reference cells on plastic surface and a positive control (10% Hydroxyethyl methacrylate (HEMA)) were tested. Mean and standard deviation are indicated. According to ISO 10993-5, samples above 70% viability are considered biocompatible. The extracts for the indirect measurements were obtained by incubating the samples for 72 h.

**Figure 3 ijms-24-11608-f003:**
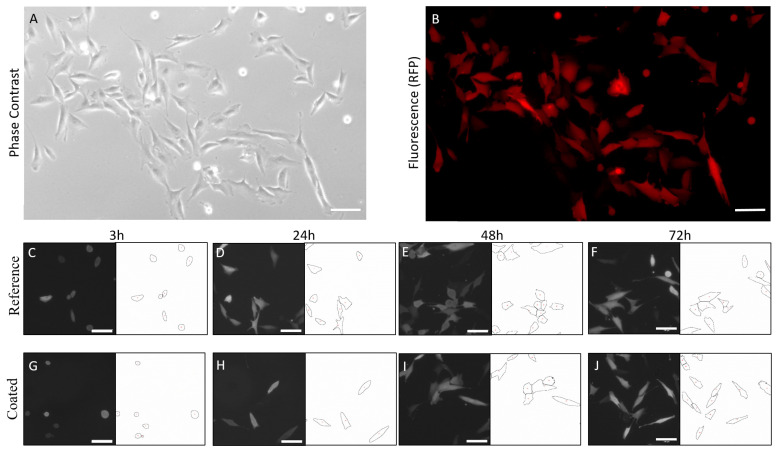
Automatic software analysis. Six phase-contrast (**A**) and fluorescence (**B**) pictures were taken from 20 reference (uncoated, **C**–**F**) and 20 coated (**G**–**J**) metal plates with MG-63-RFP cells. Image J analysis via a cell mask revealed circularity, cell area and number of cells. Scale bar = 100 µm.

**Figure 4 ijms-24-11608-f004:**
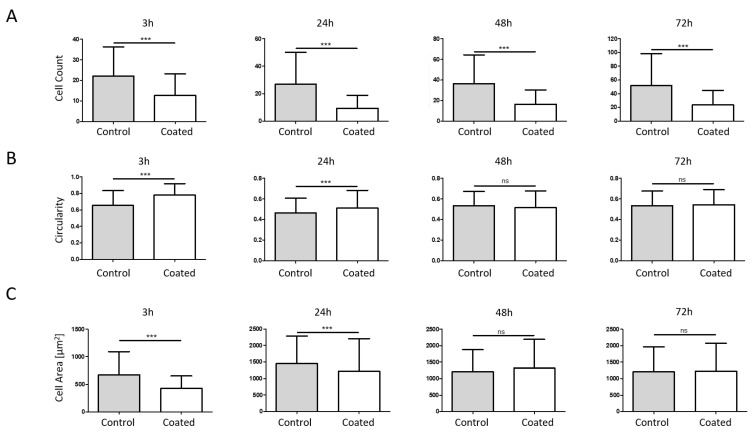
Two-dimensional phenotype analysis. Cell count (**A**), cell circularity (**B**) and cell area (**C**) of MG-63-RFP cells were analysed at four different time points (3 h, 24 h, 48 h and 72 h). Reduced cell count was observed at all time points, while circularity was increased at up to 24 h only. Single-cell area was reduced after 3 h and 24 h. (*** *p* < 0.001).

**Figure 5 ijms-24-11608-f005:**
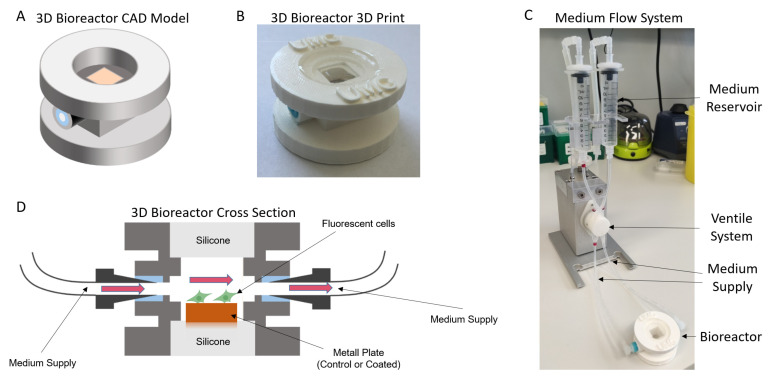
Three-dimensional bioreactor setup for investigating cell adhesion behaviour under dynamic conditions. Schematic representation of our model in CAD software (**A**) and 3D-printed model (**B**). Our bioreactor setup is attached to an Ibidi pump for flow generation inside the system (**C**). Cross-section of the bioreactor (**D**) shows the experimental setup with fluorescent cells on distinct metal plate (orange) and medium flow.

**Figure 6 ijms-24-11608-f006:**
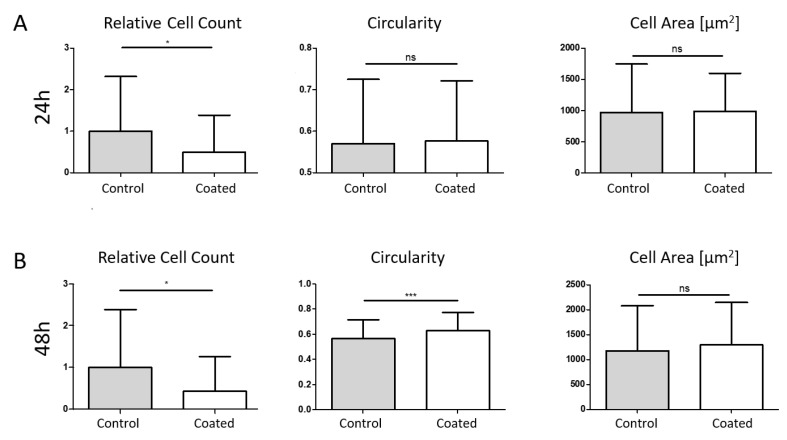
Simulation of three-dimensional and dynamic flow conditions (5 mL/min). After 24 h (**A**), cell amount on coated plates was significantly reduced while circularity and single-cell area did not change. Similar results were obtained after 48 h (**B**), where a reduced cell count was detected. Furthermore, circularity was increased while single-cell area was not changed. * *p* < 0.05, *** *p* < 0.001.

**Figure 7 ijms-24-11608-f007:**
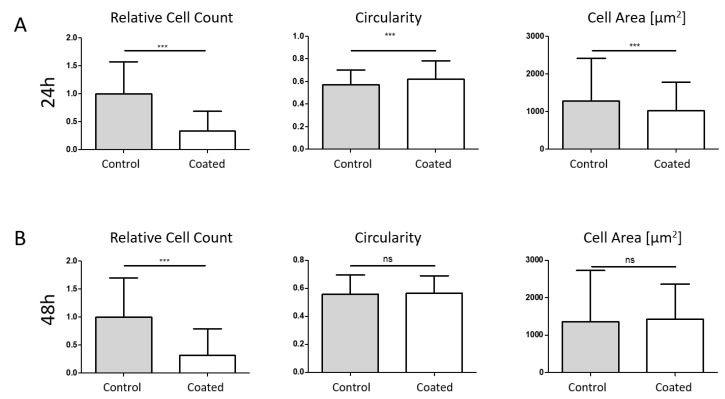
Simulation of in vivo near-three-dimensional dynamic flow conditions (10 mL/min). High flow conditions reduced cell counts after 24 h (**A**) and 48 h (**B**), while circularity was increased after 24 h and unchanged in the later time point. Furthermore, single-cell area was only reduced after 24 h. *** *p* < 0.001.

**Figure 8 ijms-24-11608-f008:**
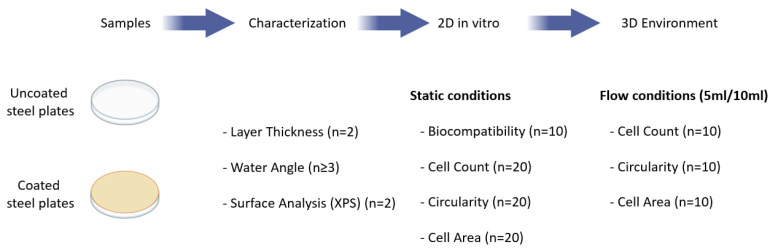
Study overview. Layer thickness, water angle and surface properties were analysed for uncoated and LightPLAS-coated medical steel plates. Both plate types were tested in 2D in vitro cell culture and biocompatibility, cell count, circularity and cell area were analysed. In the last step, both plate types were tested in a bioreactor system (3D). Cell count, circularity and cell area were measured under flow conditions, mimicking an in vivo environment.

**Figure 9 ijms-24-11608-f009:**
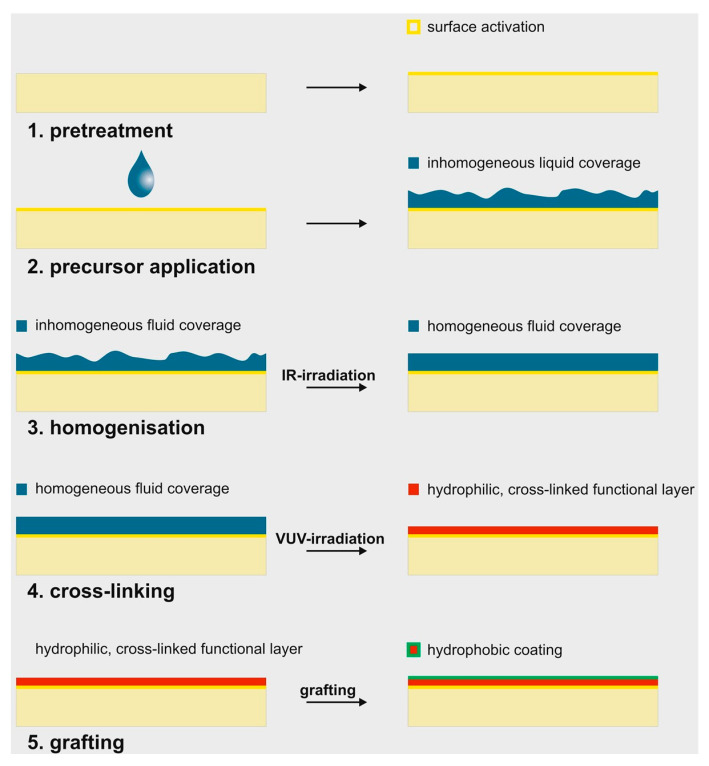
Scheme of the multi-stage coating procedure using the LightPLAS-coating technology. In the first step, the metal surface was cleaned and activated by vacuum ultraviolet radiation. Afterwards, a thin layer of PDMS was applied to the surface via aerosol application. Homogenisation of samples was achieved by IR irradiation. In the next step, VUV irradiation lead to a cross-linking on the surface, yielding a hydrophilic functional layer, which was finally grafted with HMDSO to generate a hydrophobic surface.

**Table 1 ijms-24-11608-t001:** Mean and SD of relative cell count, circularity and single-cell area of MG-63 cells cultured for 24 and 48 h in the 3D system. ns = nonsignificant, * *p* < 0.05, *** *p* < 0.001 vs. corresponding control.

		24 h		48 h	
		Mean	SD	Mean	SD
5 mL/min	Relative cell count, uncoated	100%	132%	100%	138%
	Relative cell count, coated	49% *	89%	44% *	81%
	Circularity, uncoated	0.57	0.15	0.57	0.15
	Circularity, coated	0.58 ^ns^	0.14	0.63 ***	0.14
	Single-cell area, uncoated [µm^2^]	971.0	777.1	1174.0	904.6
	Single-cell area coated [µm^2^]	989.3 ^ns^	609.4	1293.0 ^ns^	850.6
10 mL/min	Relative cell count, uncoated	100%	58%	100%	70%
	Relative cell count, coated	33% ***	36%	32% ***	47%
	Circularity, uncoated	0.57	0.13	0.55	0.14
	Circularity, coated	0.62 ***	0.16	0.56 ^ns^	0.12
	Single-cell area, uncoated [µm^2^]	1282.0	1,141,000	1355.0	1340.0
	Single-cell area, coated [µm^2^]	1032.0 ***	754,144	1426.0 ^ns^	938.7

**Table 2 ijms-24-11608-t002:** Plasmids for lentiviral production (including depositor and addgene information).

Plasmid	Depositor	Addgene Number
psPAX2	Didier Trono	#12260
pLemiR-NS	Jerry Crabtree [64]	#32809
pCMV-VSVG	Bob Weinberg [65]	#8454

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
