# Peer review of "Reduced Cell Adhesion on LightPLAS-Coated Implant Surfaces in a Three-Dimensional Bioreactor System"

_ijms, 2023, doi:10.3390/ijms241411608_

Round 1
Reviewer 1 Report
In this manuscript, the VUV light technology was used to apply a hydrophobic coating on the stainless-steel surface. The results show the effectiveness of the VUV-light technique and coatings. There are several concerns as follow.
1. As a novel technology, the coating characteristics prepared by VUV light are suggested to be investigated, such as coating bond strength, different coating thickness control, and comparison of this method to other reported methods.
2. It seems the grafted coating are responsible for the reduced cell adhesion. A detailed study on the graft coating should be provided, and also please clarify the role of the intermediate hydrophilic cross-linked functional layer, does cell have any interaction to this layer?
3. For the surface composition analysis, infrared spectrum is suggested.
Author Response
Please find the attached pdf
In this manuscript, the VUV light technology was used to apply a hydrophobic coating on the stainless-steel surface. The results show the effectiveness of the VUV-light technique and coatings.
Response: First of all, we would like to thank the Reviewer for the careful review and insightful suggestions. According to your suggestions, we have now revised the manuscript and provided a point-by-point response to each of the comments below. In the manuscript, the revised portions are highlighted in yellow.
There are several concerns as follow.
- As a novel technology, the coating characteristics prepared by VUV light are suggested to be investigated, such as coating bond strength, different coating thickness control, and comparison of this method to other reported methods.
Response: We thank the reviewer for the constructive comment. We tried to increase the information of the coating characteristics by increasing sample size for the surface energy (line 269-271).
We have added test results on layer adhesion (line 184-190, line 273-274). This is a qualitative result, not a quantitative determination of layer adhesion. Quantitative measurements are not trivial, especially on such a thin and hydrophobic surface. Such measurements are desirable and will be considered by us in the future.
Coating bond strength
Due to the low film thickness and the hydrophobic properties of the LightPLAS coating, only a qualitative test of the film adhesion to the titanium substrate could be carried out. This cross-cut test was carried out in accordance with DIN EN ISO 2409 to evaluate adhesion of a paint. A crossed grid of 6 x 6 lines with a spacing of 1 mm or 25 individual fields was manually scratched into the surface of the coating down to the stainless steel base material using a scalpel. The surface was then activated by VUV-light to enable adhesion for an adhesive tape (tesa® fabric tape 4651). The tape is peeled off and the remaining layer or the spalling near the scribe lines is evaluated. The number of investigated samples was three.
The cross-cut test for coating adhesion was GT0, i.e. no flaking or delamination was visible near the scratches after peeling off the tape.
- It seems the grafted coating are responsible for the reduced cell adhesion. A detailed study on the graft coating should be provided, and also please clarify the role of the intermediate hydrophilic cross-linked functional layer, does cell have any interaction to this layer?
Response:
We want to thank the reviewer for this important point. Exactly, the grafting is crucial for the hydrophobic properties. For us, the focus is set first on the phenomenological effects on the cells. As expected and demonstrated, cell-repellent properties are evident. The detailed analysis of the grafting is not trivial and, as you suggested, will be the next step in our view, which should then be the focus of a separate paper.
However, a detailed characterization of the grafting is necessary to understanding the interaction in detail (line 398-399).
According to our observation we think that cell interaction with the surface layer is reduced compared to uncoated plates. The rounded cell phenotype as well as the reduced cell number in our in vitro experiments indicate this lower interaction. Nevertheless, future analysis should focus on this interaction analysis. We highlighted this in the discussion part (line 434-436)
“Nevertheless, the mechanism behind the reduced cell adhesion remains unclear. Future studies should focus on the detailed analysis of surface proteins and their distribution on the cell surface. This analysis could be done by fluorescence staining and high-resolution microscopy.”
- For the surface composition analysis, infrared spectrum is suggested.
Response: We thank the reviewer for this important suggestion. We added an infrared spectrum of the coating in the results part (Supplementary Figure 1) and added the part in Material and methods as well as in the results and discussion part.
Line 192-200 (M&M part)
IR-spectroscopy
We employed infrared spectroscopy for surface composition analysis. The absorption bands exhibited characteristic chemical binding ratios, making the VUV cross-linking of the liquid silicone oil clearly visible. Infrared spectra in IRRAS (Infrared-Reflection-Absorption-Spectroscopy) configuration were obtained at an 45° angle of incidence relative to the surface normal. A single beam Fourier transform spectrometer, specifically the VERTEX 80 model from Bruker Optik GmbH, was used. The spectrometer featured a liquid nitrogen-cooled MTC detector and a resolution of 2 cm-1. To ensure precise results, we performed 128 scans and subsequently subtracted the resulting spectrum from a background spectrum obtained using a non-coated stainless steel sample as a reference. We compared the measurements of the liquid silicone oil layer after homogenization, which corresponds to step three of the coating procedure, with the final LightPLAS coating. Each analysis involved two samples (n = 2).
Figure Missing, pleaase see attached pdf
Supplementary Figure 1: IRRAS-spectroscopy for analysis of the chemical composition. Spectra of the liquid silicone layer (AK50) after homogenisation (step three) in comparison to the final LightPLAS coating i.e. after VUV cross-linking and grafting. The spectra of two individual measurements are shown in each case.
Results (line 275-277)
Supplementary Figure 1 shows the IR spectra of the non cross-linked liquid layer after the homogenisation step and the final LightPLAS coating in comparison. Two individual measurements are shown in the wavenumber range from 500 to 4000 cm-1 for each of the two variants, which are almost identical. The absorbance is given in arbitrary units.
Discussion (line 376-386)
In Supplementary Figure 1 the effect of UV cross-linking is clearly visible when looking at the absorption bands in the IR-spectra. There are several wavenumber regions where the effect of UV cross-linking is clearly visible. Of most importance is the peak of maximum absorbance of the non cross- linked silicone oil reference spectra that is located at 1111 cm-1 with a broad shoulder around 1040 cm-1. It can be assigned as an asymmetric Si-O stretching vibration. By UV-irradiation, the maximum absorbance typically is shifting to higher wavenumbers. At the same time the shoulder decreases in intensity. A further peak of relevance of the non cross-linked silicone oil can be seen at 1268 cm-1 which is assigned as symmetric bending vibration s(CH3) of methyl groups attached to Si. By UV irradiation, the symmetric bending vibration is shifted towards 1280 cm-1. Additionally, the band absorbance decreases in intensity. The shift of the s(CH3) vibration indicates a change of the structural unit from -O-Si(CH3)2-O- to -O-Si(CH3)(O-)2 caused by cross-linkage of chains as well as by the replacement of methyl by hydroxyl-groups. All observed changes in the spectra are in agreement with the data found in the literature, where further details on the different behavior of the absorption bands can be found […].
Due to the small thickness compared to the LightPLAS coating with more than 300 nm, it is not expected that the grafting produces a significant change in the IR spectrum, as the results of the IRRAS analysis always correspond to an average over the whole layer.
We want to thank the reviewer again for the careful evaluation of our manuscript. We believe that we were able to improve the quality of the manuscript thanks to your comments.

Reviewer 2 Report
Dear Authors,
The work is appropriate for the journal and the article is well written. The abstract is informative and the introduction is well composed, introducing all the implants in clinical practice and explaining the background of the technology. In addition, the authors do a good job of laying out why the coating is important in promoting or inhibiting cell adhesion. I have no comments on the abstract and introduction, both are well written.
The methods and results are also well and to my knowledge correctly written. The cell cultivation procedures are correctly presented. Personally, I have no expertise in the procedures so described in this article, especially in the coating processes, as we are working with very different materials and cells. Again, for me, the research data is credible, thorough, and well defined. However, I have no experience with the processes described in the article.
The discussion is well written and clearly.
Overall, I think the article is suitable for publication in its current form. Unfortunately, as stated, I cannot comment pertinently on the methods and results. Other parts are very good.
Author Response
Reviewer #2:
Dear Authors,
The work is appropriate for the journal and the article is well written. The abstract is informative and the introduction is well composed, introducing all the implants in clinical practice and explaining the background of the technology. In addition, the authors do a good job of laying out why the coating is important in promoting or inhibiting cell adhesion. I have no comments on the abstract and introduction, both are well written.
The methods and results are also well and to my knowledge correctly written. The cell cultivation procedures are correctly presented. Personally, I have no expertise in the procedures so described in this article, especially in the coating processes, as we are working with very different materials and cells. Again, for me, the research data is credible, thorough, and well defined. However, I have no experience with the processes described in the article.
The discussion is well written and clearly.
Overall, I think the article is suitable for publication in its current form. Unfortunately, as stated, I cannot comment pertinently on the methods and results. Other parts are very good.
Response: First of all, we would like to thank your careful review, the positive feedback for the abstract, introduction, methods and results as well as the overall positive evaluation of the manuscript. Small changes were done according to the reviewer comments. The changed parts are highlighted in yellow. We think we were able to further improve the quality of the manuscript and want to thank the reviewer for the time evaluating our manuscript.

Reviewer 3 Report
The authors overcome two difficult subjects metal implantation and surface coating of stainless steel. I have some metal pieces in my arm, so I found the article fascinating. The background explanation was well-written, and the data clearly show the effectiveness of the coating. I have one question which is what happens to the coating under mechanical rubbing against another bone or metal surface? Do you expect the coating probably to scratch rather than peel off? What are the next steps, clinical trials or experiments in animals?
Author Response
Reviewer #3:
The authors overcome two difficult subjects metal implantation and surface coating of stainless steel. I have some metal pieces in my arm, so I found the article fascinating. The background explanation was well-written, and the data clearly show the effectiveness of the coating.
Response: First of all, we would like to thank the reviewer for the careful review and overall positive evaluation of the manuscript.
I have one question which is what happens to the coating under mechanical rubbing against another bone or metal surface? Do you expect the coating probably to scratch rather than peel off?
Response: We want to thank the reviewer for this important and interesting comment. According to our tests we don´t see a peeling of the coating. Nevertheless, the coating hat some scratches after handling them in cell culture. We added these point in the discussion part (line 462-464).
“In addition, the stability of the coated surface is an important factor. In this study we didn´t observe a peeling of the surface, nevertheless scratched were observed after cell culture handling. Future studies will focus on implant and surface stability using surface analysis and biomechanics after handling.”
What are the next steps, clinical trials or experiments in animals?
Response: We want to test these surface coatings in an animal model to analyze implant ingrowth, implant stability during motion, immune response and possible influence on bone healing. We added these information in the discussion part of the manuscript (line 464-466)
“For the clinical use of such modified implants, animal models are essential to analyze implant ingrowth, implant stability during motion, immune response and possible influence on bone healing, which will be considered in future studies.”
We want to thank the reviewer again for the careful evaluation of our manuscript. We believe that we were able to improve the quality of the manuscript thanks to your comments.

Round 2
Reviewer 1 Report
The manuscript can be accepted at present form.